# The Impact of a Proprietary Blend of Yeast Cell Wall, Short-Chain Fatty Acids, and Zinc Proteinate on Growth, Nutrient Utilisation, and Endocrine Hormone Secretion in Intestinal Cell Models

**DOI:** 10.3390/ani14020238

**Published:** 2024-01-12

**Authors:** Niall Browne, Karina Horgan

**Affiliations:** Alltech Biotechnology Centre, Sarney, Summerhill Road, Dunboyne, A86 X006 Co. Meath, Ireland

**Keywords:** mouse intestinal neuroendocrine cells, porcine small intestinal cells, short-chain fatty acids, yeast cell wall, SCFAs and minerals, peptide YY, ghrelin

## Abstract

**Simple Summary:**

The production of piglets can cause stress particularly during the challenging transition of weaning them from sow’s milk to solid feed in a production setting. Supporting piglet feeding is essential to maintain overall health and growth performance, which is influenced by multiple factors. Key to driving feed intake is the release of an appetite-inducing hormone that propels animals to feed until they are full, triggering a hormone for feed cessation. Supporting the function of feed hormone release offers the potential to help piglets transition from sow’s milk to solid foods more readily. This work assessed how a combination of feed additives may potentially influence nutrient utilisation and feed hormone levels on a cellular level. The results highlight how the inclusion of a blend of key feed additives in diets may support weaning piglets through higher levels of feed hormone release and nutrient utilisation compared to traditional butyrate feed additive alone.

**Abstract:**

In piglets, it is observed that early weaning can lead to poor weight gain due to an underdeveloped gastrointestinal (GI) tract, which is unsuitable for an efficient absorption of nutrients. Short-chain fatty acids (SCFAs) such as butyrate have demonstrated their ability to improve intestinal development by increasing cell proliferation, which is vital during this transition period when the small and large intestinal tracts are rapidly growing. Previous reports on butyrate inclusion in feed demonstrated significantly increased feed intakes (FIs) and average daily gains (ADGs) during piglet weaning. Similar benefits in piglet performance have been observed with the inclusion of yeast cell wall in diets. A proprietary mix of yeast cell wall, SCFAs, and zinc proteinate (YSM) was assessed here in vitro to determine its impact on cellular growth, metabolism and appetite-associated hormones in ex vivo small intestinal pig cells and STC-1 mouse intestinal neuroendocrine cells. Intestinal cells demonstrated greater cell densities with the addition of YSM (150 ppm) compared to the control and butyrate (150 ppm) at 24 h. This coincided with the higher utilisation of both protein and glucose from the media of intestinal cells receiving YSM. Ghrelin (an appetite-inducing hormone) demonstrated elevated levels in the YSM-treated cells on a protein and gene expression level compared to the cells receiving butyrate and the control, while satiety hormone peptide YY protein levels were lower in the cells receiving YSM compared to the control and butyrate-treated cells across each time point. Higher levels of ghrelin and lower PYY secretion in cells receiving YSM may drive the uptake of protein and glucose, which is potentially facilitated by elevated gene transporters for protein and glucose. Greater ghrelin levels observed with the inclusion of YSM may contribute to higher cell densities that could support pig performance to a greater extent than butyrate alone.

## 1. Introduction

In piglets, early weaning can lead to poor weight gain as a result of an underdeveloped GI tract that is unsuitable for the efficient absorption of nutrients. The central melanocortin system is key to regulating feed intake (FI), metabolic activities, and energy homeostasis [1,2], with the regulation of energy metabolism disseminated by hormone release. Key to appetite regulation are hormones that drive appetite, such as growth hormone (GH)-releasing peptide (ghrelin), and hormones that supress appetite, including leptin, peptide YY (PYY), and cholecystokinin. The appetite suppressant PYY decreases food intake and reduces weight gain, as observed when PYY is administered to mice [3]. Conversely, ghrelin levels rise prior to feeding and fall quickly after ingestion of nutrients. Ghrelin, a circulating orexigen, is mainly produced by the endocrine cells of the gastric mucosa and, to a lesser extent, in the small intestine [4]. Despite a wide range of studies being conducted to explain the action of ghrelin in animals, in terms of biological functions and its mechanism of action, it is not clearly understood. It is surmised that ghrelin plays an important role in feed efficiency because it is used to maintain energy balance [5].

The intestinal lumen of animals contains a range of short-chain fatty acids (SCFAs) that are mainly produced from bacterial fermentation of indigestible fibres that aid the production of acetate, butyrate, and propionate [6]. Of these, butyrate has demonstrated an ability to improve intestinal development by increasing cell proliferation, which is vital during transition periods when the small and large intestinal tract are rapidly growing [7]. Piva et al. [8] demonstrated that pigs had significantly increased daily feed intakes with the addition of sodium butyrate during weaning from 15 to 35 days after birth. Average daily gains (ADGs) were also significantly greater after 14 days of treatment (800 mg/kg of diet). Although these initial gains are promising, they do not continue past 35 days of age, which may suggest that the initial advantages level out after the intestinal development is achieved in those without sodium butyrate treatment.

The addition of mannan-rich prebiotics to piglet feed has led to several benefits in weight gain, feed conversion, and intestinal health while decreasing pathogen load and mortality rates [9,10]. Furthermore, the addition of mannan oligosaccharides (MOSs) to poultry diets has also improved villi and crypt structure, which in turn attributed to greater energy absorption through maximising the intestinal surface absorption area for nutrients [11,12,13]. Similar benefits in morphological changes have also been observed in nursery piglets receiving diets containing MOSs [14]. Similarly extensive benefits have also been observed with mineral zinc additions to pig diets regarding intestinal health, performance [15], and repair [16].

Weaning piglets are more susceptible to infections and have reduced appetites, which can lead to overall poor performance [17]. The benefits observed in animals receiving both yeast MOSs, SCFAs, and mineral zinc in their diets has led to the development of a proprietary blend that includes a yeast cell wall, SCFAs, and zinc proteinate (YSM). YSM’s inclusion in diets was evaluated in nursery pigs between 8 and 13 kg, with YSM supplementation leading to significantly greater FI, ADGs, and body weight (BW) compared to control pigs [18]. This study aimed to determine if YSM alters feed intake by assessing its impact on appetite hormone release from the intestinal tract in vitro using a mouse neuroendocrine cell line and a primary pig intestinal cells, both of which possess many features of native intestinal endocrine cells. The proposed benefits of increased appetite driven by endocrine secretions, combined with optimal gut function, enhanced metabolism, and a promotion of cellular growth, was compared to the industry standard sodium butyrate.

## 2. Materials and Methods

An enzymatic digestion was performed on 0.5 g of both YSM (Viligen, obtained from Alltech Inc., Nicholasville, Kentucky) and the treatment control, sodium butyrate (Sigma Aldrich, Burlington, MA, USA), to mimic gastric digestion. Digestion was adapted from a 3-step in vitro technique by Boisen and Fernandez [19], including pepsin digestion carried out at pH 2 for 2 h at 39 °C followed by a pancreatin digestion at pH 6.8 for 4 h at 39 °C, with experimental volumes outlined by Meunier et al. [20]. A post-digestion concentration of 150 ppm of either sodium butyrate or YSM was used to treat cells.

### 2.1. Cell Culture

Two mammalian cell models were used that were capable of feed hormone expression. Porcine small intestinal cells (pig cells) (Creative Bioarray, Shirley, NY, USA, CSC-C8659W) and mouse intestinal neuroendocrine cells, STC-1 (ATCC^®^ CRL-3254™, Dongguan, China) cells, were used for these experiments. Cells were grown in sterile Dulbecco’s MEM high-glucose media supplemented with 10% (*v*/*v*) foetal bovine serum (FBS) and 1% (*v*/*v*) L-glutamine Gibco^®^ (Grand Island, NY, USA), and a specific Creative Bioarray L-glutamine and FBS supplied for the pig intestinal cells. The cells were incubated at 37 °C with a 5% CO_2_ atmosphere. Pig intestinal cells (Passage 1–5) or STC-1 (Passage 2–15) cells were seeded onto T25 flasks at a density of 0.5 × 10^6^ cells for analysing the fixed effects of each treatment (YSM, butyrate, and control), at each time point (24, 48, and 72 h), and at three independent occasions as represented by different cell passages. Cells were left to attach to the surface overnight at 37 °C in a 5% CO_2_ atmosphere prior to use and mycoplasma testing was also carried out prior to use (AssayGenie, Dublin, Ireland, MORV0011).

The T25 cell culture flasks were seeded at 1 × 10^5^ cells and allowed to adhere prior to independent flasks receiving either butyrate, YSM, or no treatment for the control flask, for both pig intestinal cells and STC-1 cells, and left for either 24, 48, or 72 h before analysis. Cells were lysed for RNA expression analysis and media were collected for measuring the glucose and/or protein abundance for both cell lines. Cells were enumerated following trypsinization for pig intestinal cells and mouse cells at the respective time points using haemocytometer.

### 2.2. Isolation of Cell-Free Media and Cellular Lysate Components

Cell-free media were collected in 15 mL centrifuge tubes from the T25 flasks at the designated time point and centrifuged at 1500 rpm to pellet cells and cellular debris that may compromise the analysis of protein levels, glucose measurement, and ELISA measurements.

### 2.3. Glucose Analysis

Pig intestinal cell-free media glucose levels were determined following incubation of the cells in the presence of butyrate or YSM or without treatment (control) for 0, 24, 48, or 72 h with a glucose assay kit from Sigma Aldrich (GAGO-20). Cell-free media (500 µL) were used to determine the glucose concentration in each treatment group according to the manufacturer’s instructions. A 0 h reading of each treatment group was used as the starting glucose concentration with subsequent time points subtracted from 0 h to determine the glucose levels utilised from the media.

### 2.4. Protein Analysis

A three-point calibration curve was performed according to the manufacturer’s instructions using the Qubit™ Standard 1–3 reagents provided. The cell-free media were measured for the total protein concentration in triplicate by combining 10 µL of cell-free media with 190 µL Qubit™ working solution and measuring on a Qubit^®^Fluorometer (Thermo Fisher Scientific, Waltham, MA, USA). Samples from 0 h, 24 h, 48 h, and 72 h were analysed and protein concentrations were subtracted from their respective starting protein concentrations at 0 h for each treatment group to determine the protein usage from the media.

### 2.5. ELISA Measurement

ELISAs from Sigma and AssayGenie were performed to measure secreted hormones PYY (RAB0413-1KT, mouse) (PREB0269, pig) and ghrelin (RAB0207-1KT, mouse) (PRFI00246, pig). Cell-free media (100 µL) from each set of treated cells were used to determine peptide hormone concentrations using specific ELISA analytical methods.

### 2.6. RNA Isolation

Cells grown in the presence of butyrate or YSM, or their absence for the control, were suspended in 350 µL of RLT lysis buffer (Qiagen, Hilden, Germany). The cell suspension was ruptured for 30 s with a tissue rupture probe on ice to generate a homogenous mixture. RNA was isolated following the RNeasy Micro Kit (Qiagen, Hilden, Germany) procedure as detailed in the user manual. RNA quality was determined using a Qubit 4 Fluorometer (Invitrogen, Waltham, MA, USA) with IQ values above 6.5, and RNA concentrations ˃ 1.8 μg were used for qPCR cDNA synthesis (SuperScript^®^-III).

### 2.7. Gene Expression Analysis

Reverse transcription was performed using RNA as a template to synthesise cDNA using a SuperScript^®^ III First-Strand Synthesis System (Invitrogen, USA) prior to PCR analysis. Gene expression changes associated with real-time PCR were measured in triplicate using a two-step cycling programme, consisting of a heat activation step (95 °C for 10 min) and a cycling step (40 cycles, 95 °C for 15 s and 60 °C for 1 min) (ABI 7500 Fast; Applied Biosystems, Waltham, MA, USA).

PCR analysis of gene expression changes with associated markers for metabolism and endocrine expression was performed using the custom-designed PCR primers for the pig intestinal cells, *EAAT*, *CAT-1*, *PYY*, *ghrelin*, *Glut-2*, and *GAPDH*, and for the STC-1 cells, *EAAT*, *CAT-1*, *PYY*, *ghrelin*, *Glut-2*, and *Actin* (Table 1). Differential expression results were determined by comparing the cells treated with butyrate and YSM to the untreated cells.

### 2.8. Statistical Analysis

Three independent biological replicates were performed, with statistical analyses of all experiments analysed via one-way ANOVA, Tukey’s post hoc test with significance determined at *p* ≤ 0.05, and Graph Pad PRISM 5.0. Bar charts are represented as the mean average and standard error mean (SEM).

## 3. Results

### 3.1. Pig Intestinal Cell and STC-1 Intestinal Cell Density

The pig intestinal cells receiving YSM (150 ppm) demonstrated significantly greater cell densities (2.05 × 10^5^ ± 1.83 × 10^4^ CFU/mL) compared to the control (1.23 × 10^5^ ± 1.87 × 10^4^ CFU/mL, *p* ≤ 0.05) and butyrate (150 ppm)-treated cells, which had lower cell densities (1.11 × 10^5^ ± 0.42 × 10^4^ CFU/mL, *p* ≤ 0.01) at 24 h (Figure 1A). A similar trend was observed at 48 h for the butyrate-treated cells, which had lower cell densities (3.1 × 10^5^ ± 2.15 × 10^4^ CFU/mL, *p* ≤ 0.05) compared to the cells receiving YSM (4.2 × 10^5^ ± 5.01 × 10^4^ CFU/mL). The 72 h time point demonstrated significantly lower cell concentrations in the butyrate-treated cells (5.13 × 10^5^ ± 2.42 × 10^4^ CFU/mL) compared to the control (6.32 × 10^5^ ± 1.12 × 10^4^ CFU/mL, *p* ≤ 0.05) and the cells receiving YSM (7.08 × 10^5^ ± 3.63 × 10^4^ CFU/mL, *p* ≤ 0.01).

The STC-1 mouse intestinal cells capable of endocrine secretions were found to have significantly higher cell densities at 24 h when treated with YSM (150 ppm) (2.06 × 10^6^ ± 1.84 × 10^5^ CFU/mL) compared to both the control (1.27 × 10^6^ ± 2.64 × 10^5^ CFU/mL, *p* ≤ 0.001) and butyrate-treated cells (1.57 × 10^6^ ± 2.09 × 10^5^ CFU/mL, *p* ≤ 0.001) (Figure 1B). Similarly, at 48 h it was observed that cells receiving YSM (2.99 × 10^6^ ± 4.19 × 10^5^ CFU/mL) had a significantly greater cell concentration over the butyrate-treated cells (2.10 × 10^6^ ± 4.11 × 10^5^ CFU/mL, *p* ≤ 0.001) and the control cells (1.96 × 10^6^ ± 3.06 × 10^5^ CFU/mL, *p* ≤ 0.001). Cell density was also significantly higher at 72 h for the cells receiving YSM (4.67 × 10^6^ ± 1.31 × 10^5^ CFU/mL, *p* ≤ 0.01) compared to the butyrate-treated cells (2.72 × 10^6^ ± 7.26 × 10^5^ CFU/mL).

### 3.2. Pig and STC-1 Intestinal Cell Media Glucose Levels

The addition of YSM to the pig intestinal cells coincided with a significantly greater glucose utilisation (12.64 ± 0.22 µg/mL) at 24 h compared to the control (11.63 ± 0.68 µg/mL, *p* ≤ 0.001). Similarly, at 48 h, the cells receiving YSM (13.08 ± 0.53, *p* ≤ 0.05) demonstrated significantly greater glucose usage compared to the control cells (11.71 ± 0.81 µg/mL), but no significant difference was observed in the butyrate-treated cells at 24 and 48 h. (Figure 2A). Significantly greater glucose usage was also found in the cells receiving YSM (14.44 ± 1.62) compared to the cells treated with butyrate (12.2 ± 0.58 µg/mL, *p* ≤ 0.05) or the control cells (12.18 ± 0.72 µg/mL, *p* ≤ 0.05) at 72 h.

Glucose utilisation in the media was significantly greater in STC-1 cells receiving YSM (12.94 ± 0.33 µg/mL) at 24 h compared to the control (2.27 ± 2.09 µg/mL, *p* ≤ 0.01) and butyrate-treated cells (4.16 ± 0.67 µg/mL, *p* ≤ 0.01). The control cells’ usage of glucose from the media at 48 h was significantly lower (1.96 ± 2.96 µg/mL, *p* ≤ 0.05) compared to the cells receiving YSM (2.27 ± 2.09 µg/mL). At 72 h, both the control and butyrate-treated cells exhibited a greater utilisation of glucose compared to their respective 48 h time point; despite this, for the butyrate-treated cells (7.55 ± 0.90 µg/mL, *p* ≤ 0.01) and the control (15.04 ± 0.87 µg/mL, *p* ≤ 0.05), glucose utilisation was still significantly lower at 72 h when compared to the cells receiving YSM (15.52 ± 1.65 µg/mL) (Figure 2B).

### 3.3. Protein Usage by Pig and STC-1 Mouse Intestinal Cells

The protein utilised from the media was significantly higher in pig intestinal cells receiving YSM at the 48 h (255.33 ± 10.52 µg/µL, *p* ≤ 0.05) and 72 h (328.33 ± 9.14 µg/µL, *p* ≤ 0.05) time points compared to the control. Lower protein utilisation was seen in the cells receiving butyrate at the 48 h (236.33 ± 13.79 µg/µL) and 72 h (321.33 ± 5.88 µg/µL) time points compared to the YSM-treated cells (Figure 3A). Similar changes in protein usage were observed in the STC-1 mouse intestinal cells, with a significantly greater usage observed in the cells receiving YSM (106.66 ± 8.0 µg/µL, *p* ≤ 0.05) compared to the control (57.33 ± 20.17 µg/µL) at 48 h (Figure 3B).

### 3.4. Endocrine Hormone Secretion from Pig and STC-1 Intestinal Cells

The concentration of PYY in the pig intestinal cells receiving YSM was significantly lower than the control at 72 h, with minimal changes observed between 24 h and 48 h (Figure 4A). The mouse STC-1 intestinal cells’ abundance of PYY was significantly lower at 48 h in both the YSM- (0.79 ± 0.05 ng/µL, *p* ≤ 0.05) and butyrate-treated cells (0.83 ± 0.04 ng/µL, *p* ≤ 0.05) compared to the control (0.95 ± 0.03 ng/µL), although no significant differences were observed at 24 h (Figure 4B).

The YSM addition (0.74 ± 0.00018 ng/µL) caused a non-significant rise in ghrelin over the control (0.74 ± 8.62^−5^ ng/µL) and a significant rise compared to the butyrate-treated (0.73 ± 2.02^−4^ ng/µL, *p* ≤ 0.05) pig intestinal cells at 48 h (Figure 4C). A significantly lower level of ghrelin secretion from the cells receiving butyrate (0.7403 ± 1.18^−4^ ng/µL, *p* ≤ 0.05) was detected at 72 h compared to the control (0.7416 ± 2.32^−4^ ng/µL). In the STC-1 cells, the concentration of ghrelin was significantly greater in the control cells (1.25 ± 0.039 ng/µL, *p* ≤ 0.05) at 24 h over the cells receiving YSM (1.177 ± 0.024 ng/µL). Although, at 48 h, ghrelin hormone levels were significantly greater in the media from the cells receiving YSM (1.319 ± 0.010 ng/µL, *p* ≤ 0.05) compared to the control cells (1.244 ± 0.025 ng/µL) (Figure 4D).

### 3.5. Endocrine Gene Expression from Pig and STC-1 Intestinal Cells

Appetite-suppressing hormone *PYY* from pig intestinal cells demonstrated significantly elevated expression levels in the cells receiving butyrate at 24 h (6.26 ± 0.15) compared to the control (1.04 ± 0.10, *p* ≤ 0.001) and the cells receiving YSM (1.35 ± 0.04, *p* ≤ 0.001) (Figure 5A). Similarly, cells receiving butyrate (1.04 ± 0.10, *p* ≤ 0.05) at 72 h also had a significant elevation in their *PYY* expression over the control (1.39 ± 0.19).

The STC-1 cells’ *PYY* (Figure 5B) expression was significantly higher in STC-1 cells receiving butyrate (41.4 ± 9.77) at 24 h compared to the control (2.57 ± 1.02, *p* ≤ 0.01) and the cells receiving YSM (12.95 ± 2.45, *p* ≤ 0.05). *PYY* levels were significantly higher in both the YSM-treated (2.60 ± 1.20, *p* ≤ 0.001) and butyrate-treated (11.22 ± 2.43, *p* ≤ 0.05) cells compared to the control (0.70 ± 0.111), with the butyrate addition leading to the greatest concentration of *PYY* at 48 h.

*Ghrelin* gene expression in pig intestinal cells was significantly greater in the control at 24 h (1.36 ± 0.79) and 48 h (0.95 ± 0.03) compared to the YSM-exposed cells (0.04 ± 0.03, *p* ≤ 0.01 and 0.92 ± 0.05, *p* ≤ 0.001) and the butyrate-supplemented cells (0.25 ± 0.06, *p* ≤ 0.05 and 0.57 ± 0.06, *p* ≤ 0.001) for the same respective time points (Figure 5C). Although, a significantly greater expression of *ghrelin* was observed at 72 h in the YSM-supplemented cells (3.73 ± 1.02, *p* ≤ 0.01) compared to the control (1.28 ± 1.02) and, to a lesser extent, in the butyrate-treated cells (3.03 ± 0.61, *p* ≤ 0.05) compared to the control.

*Ghrelin* gene expression was also significantly greater in STC-1 cells receiving YSM (1.59 ± 0.06, *p* ≤ 0.01) compared to the control cells (1.02 ± 0.02) at 24 h. Similarly, greater ghrelin expression was observed in the YSM-treated cells at 48 h (44.07 ± 19.35, *p* ≤ 0.05) compared to the control cells (2.60493 ± 1.20), while the butyrate-treated cells (6.47 ± 2.26) demonstrated no significant difference from the control (Figure 5D).

### 3.6. Gene Expression of Glucose Transporter Glut-2 from Pig and STC-1 Intestinal Cells

*Glut-2* gene expression in pig intestinal cells was demonstrated to be significantly higher at 24 h for the YSM treatment (1.76 ± 1.01) over the control (0.69 ± 0.20, *p* ≤ 0.01) and butyrate-treated (1.26 ± 0.73, *p* ≤ 0.05) cells. Butyrate-treated cells had a significantly higher *Glut-2* expression at 72 h (1.75 ± 0.48) over the control (0.29 ± 0.24, *p* ≤ 0.01) and YSM-treated cells (0.11 ± 0.20, *p* ≤ 0.01) (Figure 6A).

*Glut-2* also demonstrated significantly greater gene expression at 24 h (1.36 ± 0.09, *p* ≤ 0.05) and 48 h (1.81 ± 0.15, *p* ≤ 0.05) for the STC-1 intestinal cells receiving YSM compared to the control group at 24 h (0.75 ± 0.16) and 48 h (0.78 ± 0.08), respectively (Figure 6B). No significant difference in expression was seen between the butyrate- (1.26 ± 0.28) and YSM-treated STC-1 cells (Figure 6B).

### 3.7. Pig and STC-1 Intestinal Cell Gene Expression of Amino Acid Transporters EAAT-3 and CAT-1

The expression of metabolic genes excitatory amino acid transporter 3 (*EAAT-3*) and cationic amino acid transporter-1 (*CAT-1*) was assessed in the butyrate-, YSM-, and untreated control cells to establish if metabolism was altered. *EAAT-3* expression was significantly lower in the butyrate-treated pig cells (0.03 ± 0.02, *p* ≤ 0.05) compared to the control (0.38 ± 0.27 (Figure 7A). STC-1 cells receiving butyrate demonstrated a significantly higher *EAAT-3* (2.82 ± 0.31, *p* ≤ 0.001) expression compared to the control (1.03 ± 0.08) and YSM-treated (1.26 ± 0.06) cells, although YSM exhibited a significantly greater expression of *EAAT-3* (27.9 ± 5.01) compared to the control (2.31 ± 0.96, *p* ≤ 0.05) and butyrate-treated group (3.25 ± 0.58) at 48 h (Figure 7B).

*CAT-1* expression in the pig intestinal cells was significantly higher following YSM addition after 48 h (1.57 ± 0.28) and 72 h (1.17 ± 0.05) compared to the control at 48 h (0.90 ± 0.16, *p* ≤ 0.001) and 72 h (0.97 ± 0.07, *p* ≤ 0.01), respectively, and the butyrate-treated cells at 72 h (0.84 ± 0.18, *p* ≤ 0.001) (Figure 7C). Similarly, *CAT-1* expression in STC-1 cells was significantly greater in the cells receiving YSM at 24 h (1.64 ± 0.04) compared to the control (1.06 ± 0.07, *p* ≤ 0.001) and butyrate-treated cells (1.32 ± 0.23, *p* ≤ 0.05) (Figure 7D). STC-1 cells had a similar significant difference in their gene expression of *CAT-1*, which was higher in the YSM-treated cells (21.34 ± 3.45) than in the control (2.47 ± 0.92, *p* ≤ 0.05) and butyrate-treated cells (0.44 ± 0.07, *p* ≤ 0.05) at 48h (Figure 7D).

## 4. Discussion

The promotion of piglet gut health is of critical importance to ensure a strong start during weaning. A piglet’s performance is influenced strongly by an underdeveloped immune system and digestive tract, which attributes to greater infections that lead to post-weaning diarrhoea (PWD) and, ultimately, impaired growth. A widescale approach to promoting pig growth was previously achieved by using growth promoters such as antibiotics, which indiscriminately removed both beneficial bacteria and Gram-negative bacteria that cause PWD. But due to the growth of resistant bacteria, this practice was stopped in 2005 in Europe (European Commission 2005) and restricted heavily in the United States of America by the Food and Drug Association 2012 [25].

Feed intake by weanlings is vital for their growth and development, with levels of ghrelin within the gut of piglets being key to driving feeding. This is demonstrated by the addition of ghrelin in animals having a strong correlation to food intake in a dose-dependent manner, particularly in younger animals who respond more effectively and intensely to ghrelin [26]. Furthermore, higher levels of ghrelin are linked to greater ADGs when exogenous ghrelin is administered to weaning piglets [27]. While ghrelin’s release is strongly influenced by the method of piglet feeding with set feed times causing a spike in ghrelin before feeding, Adlib feeding does not generate ghrelin spikes [28]. To this end SCFAs such as butyrate offer a potential solution to the issue of piglet gut underdevelopment as butyrate is a freely absorbed energy source to enterocytes [29,30], which rapidly induces intestinal development and function.

The potential impact of YSM was assessed here utilising STC-1 endocrine intestinal cells and primary pig intestinal cells that are both capable of producing PYY and ghrelin hormones. Their response to supplementation was assessed by measuring nutrient utilisation through protein and glucose abundance in media. ELISA was also employed to measure PYY and ghrelin abundance and their respective gene expression levels were determined.

The addition of YSM to pig and mouse cells derived from the small intestines resulted in significantly greater cell densities compared to control cells and the butyrate-treated cells for up to 72 h (Figure 1A,B). Higher densities of cells is an indicator of increased cell growth potential. Greater cell proliferation also coincided with a greater usage of glucose (Figure 2A,B). YSM has previously been demonstrated to improve growth performance in both average daily gains and overall weight gain with its addition to nursery pig feed [18]. The beneficial impact on cellular proliferation and development has been highlighted in response to SCFAs (7) and similarly from the inclusion of prebiotics in the form of yeast mannan [31]. The combined effect of yeast mannan and SCFAs in YSM may support better cellular growth performance over relying on SCFAs alone. Evidence for prebiotics improving overall growth has been observed in monogastric species including pigs [32,33] and chickens [34,35].

Furthermore, cellular use of glucose is a strong measure of mammalian cell growth [36]; and for this reason, it was assessed here to determine if differences arising in cell growth were associated with greater consumption of glucose. Primary pig cells receiving YSM demonstrated a greater cellular glucose consumption compared to the control at 24 h, 48 h, and 72 h (Figure 2A). Butyrate-treated cells demonstrated a significantly lower usage of glucose compared to the cells receiving YSM at 24 h and 72 h. This was similarly mirrored for the STC-1 cells (Figure 2B), highlighting that both cell lines supplemented with YSM had higher uptakes of glucose from the media when compared to the control and, to a lesser extent, the butyrate-treated cells.

The Glut-2 transporter is responsible for regulating glucose transportation and uptake in cells. The expression of the gene for this transporter was greater in the pig intestinal cells that received YSM at 24 h (Figure 6A) when compared to the control and butyrate-treated cells. This coincided with the higher *Glut-2* expression observed in STC-1 cells (Figure 6B) at 24 h, although this was also observed at the 48 h time point, unlike the pig intestinal cell expression of *Glut-2*, which was highest at 24 h. The pig intestinal cells had significantly greater levels of *Glut-2* expression in their butyrate-treated cells, which may have resulted from a delayed response for sustaining growth requirements compared to the cells receiving YSM, which peaked at 24 h, and the control that peaked at 48 h. Cell densities observed in the YSM-treated cells were also elevated over both the control and butyrate-treated cell groups. The elevated expression of *Glut-2* may have contributed to the greater utilisation of glucose from the media of cells receiving YSM (Figure 2A,B).

Drozdowski et al. [37] found that the inclusion of SCFAs increased *Glut-2* expression but that butyrate on its own was not sufficient to increase *Glut-2* expression, which is possibly why the butyrate treatment on its own here did not elicit the same increase as the cells receiving YSM. This is similarly mirrored in the STC-1 cells, with a significant increase over the control but not in the butyrate-treated cells (Figure 6B). The same research group also found that glucose consumption was increased with the inclusion of SCFAs, which was similarly observed here and may contribute to the significantly improved growth, particularly in the YSM-treated cells compared to the control. Comparably, an earlier study that continuously fed SCFAs to rats reported increased mRNA and protein abundance of Glut-2 within 6 h [38].

An important parameter of cellular function and growth is the uptake of amino acids, which was assessed here using a crude measurement of the overall protein content in media from the three groups receiving butyrate, YSM, or no treatment. Greater protein use was apparent in both the STC-1 cells and pig intestinal cells following YSM application, as determined by less overall protein content from cell media. Several studies on yeast mannan have highlighted its benefits in increasing amino acid transport across the intestinal tract [39,40], which has been attributed to greater expression of amino acid transporters [41].

The consumption rate of food is strongly regulated by hormones that supress appetite, including PYY, and those that drive appetite, such as ghrelin; for this reason, STC-1 cells and pig intestinal cells were assessed here for their expression of these appetite-regulatory hormones. Greater gene expression levels of *PYY* (Figure 5A,B) were observed in response to butyrate over the control. Several publications have highlighted that in vitro models receiving butyrate had increased *PYY* expression [42,43] and PYY hormone secretion [44]. The pig intestinal cells receiving YSM demonstrated a significantly lower abundance of PYY compared to the control at 72 h (Figure 4A). Secreted PYY hormone levels were similarly high in the butyrate-treated cells at 24 h, but a significantly lower PYY abundance was found in the butyrate- and YSM-treated cells at 48 h compared to the control. This may be attributed to time differences in protein translation from gene expression and to the time it takes for proteins to be absorbed into the cells.

On the other hand, ghrelin hormone production appeared to peak after 48 h in both the pig cells and mouse STC-1 cells. A significantly greater secretion of ghrelin was seen in the STC-1 cells receiving YSM at 48 h when compared to the control (Figure 4D) and in the pig intestinal cells in comparison to the butyrate-treated cells (Figure 4C). An in vitro assessment of SCFAs demonstrated a significant increase in ghrelin in the supernatant of cells in response to all SCFAs: acetate, propionate, and butyrate [44]. An earlier study reported increasing levels of *ghrelin* expression from gastric cells in response to butyrate [45], which may account for the higher expression observed in the YSM- and, to a lesser extent, butyrate-treated cells over the control. The greater abundance of ghrelin in response to the addition of YSM also corroborates what has been demonstrated in nursey pig trials, which found higher feed intake rates with the inclusion of YSM in feed [18]. Interestingly, *ghrelin* gene expression peaked at 72 h in the pig intestinal cells and 48 h in the STC-1 intestinal cells, which may be attributed to the differences in expression responses to stimuli across the cell models.

Another driver of growth and metabolism is linked strongly to mTOR activity, with arginine highlighted as an important amino acid in regulating cellular metabolism through its ability to activate the mTORC1 kinase [46,47]. One key amino acid transporter, CAT-1, supplies cationic amino acids for cellular metabolism, especially arginine [48]. Similarly, EAAT-3 has been shown to facilitate the transport of anionic amino acids that activate the mTOR pathway, which supports pig intestinal cell proliferation [49]. Arginine transported by CAT-1 and glutamate transported through EAAT-3 are linked separately to the regulation and activation of mTOR. Both *CAT-1* and *EAAT-3* gene expression levels were greater at later time points in STC-1 and pig intestinal cells that received YSM, which may influence mTOR activation.

Taken together, the elevated cell densities may be driven by higher ghrelin levels that in turn raise the cellular metabolism of nutrients, which include glucose and protein from the media in cells receiving YSM. With SCFA supplementation, the effects on growth performance are varied depending on the concentrations and combinations of SCFAs used [42,43,44].

## 5. Conclusions

Our study demonstrated that YSM promoted cellular growth through greater consumption of glucose facilitated by glucose transport through Glut-2, in combination with higher protein usage, which was potentially supported by greater appetite-associated hormone secretion. These in vitro findings demonstrate that YSM may have the potential to improve the feed consumption of weaning pigs. Enhancing appetite in weaning piglets is vital to ensuring the sustained growth and development of young pigs.

## Figures and Tables

**Figure 1 animals-14-00238-f001:**
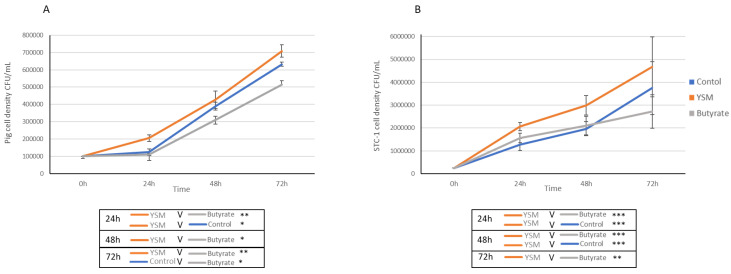
Monitoring of cell densities of pig (**A**) and mouse (**B**) intestinal cells supplemented with YSM and butyrate, and the un-supplemented control. Statistical analyses using one-way ANOVA and Tukey’s post hoc test (*n* = 3), *p* ≤ 0.05 *, *p* ≤ 0.01 **, and *p* ≤ 0.01 ***, are presented in the table below respective graphs (**A**) and (**B**).

**Figure 2 animals-14-00238-f002:**
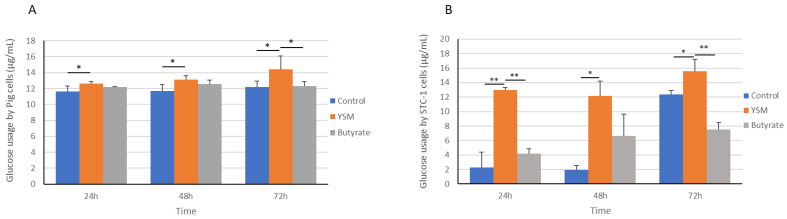
Glucose utilised from media by pig (**A**) and STC-1 intestinal cells (**B**) in response to supplementing with butyrate or YSM, or the non-supplemented control, for 24, 48, and 72 h. Statistical analysis was performed using one-way ANOVA and Tukey’s post hoc test (*n* = 3), *p* ≤ 0.05 *, *p* ≤ 0.01 **.

**Figure 3 animals-14-00238-f003:**
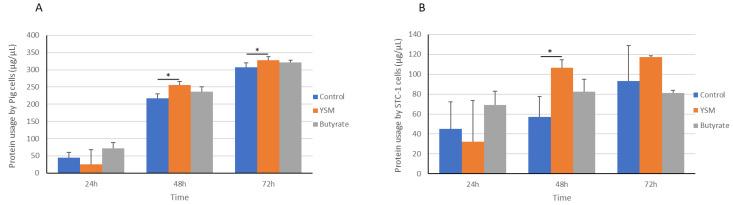
Protein usage from media by pig (**A**) or STC-1 mouse intestinal cells (**B**) in response to supplementing with butyrate or YSM, or the non-supplemented control, for 24, 48, and 72 h. Statistical analysis was performed using one-way ANOVA and Tukey’s post hoc test (*n* = 3), *p* ≤ 0.05 *.

**Figure 4 animals-14-00238-f004:**
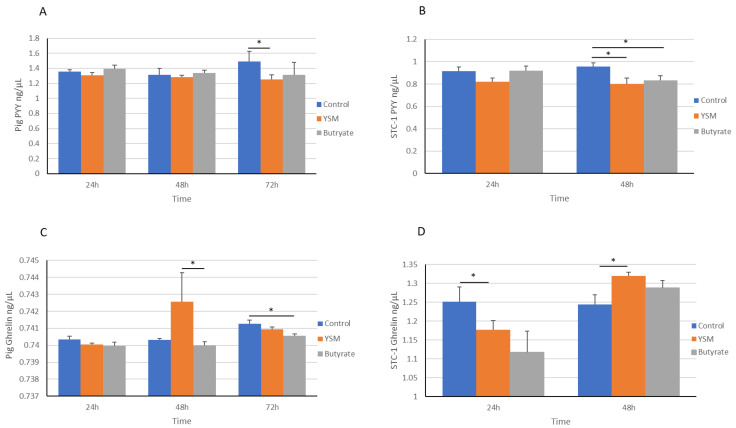
ELISA analysis of intestinal cell levels of the satiety protein PYY in pig (**A**) and STC-1 cells (**B**) and appetite inducer ghrelin abundance in pig (**C**) and mouse cells (**D**) supplemented with YSM or butyrate or the non-supplemented control. Statistical analysis was performed using one-way ANOVA and Tukey’s Post hoc test (*n* = 3), *p* ≤ 0.05 *.

**Figure 5 animals-14-00238-f005:**
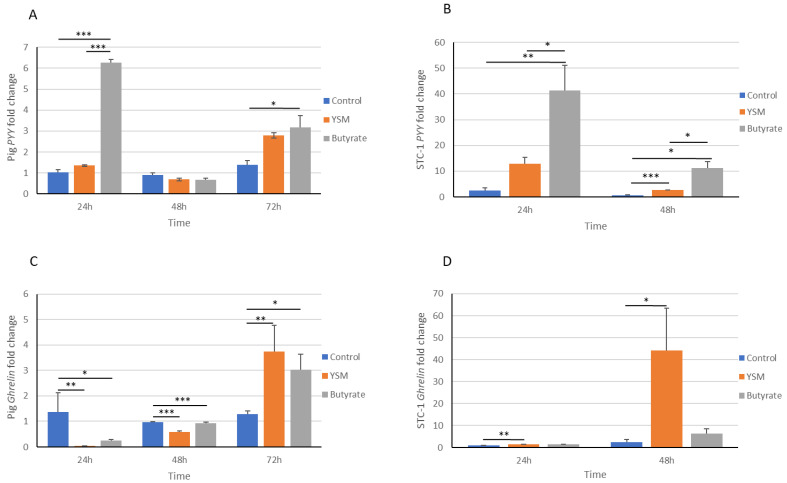
Gene expression analysis of the effects of butyrate and YSM on intestinal cell levels of the satiety gene *PYY* in pig (**A**) and mouse (**B**) cells and the appetite inducer *ghrelin* in pig (**C**) and mouse (**D**) cells. Statistical analysis was performed using one-way ANOVA and Tukey’s post hoc test (*n* = 3), *p* ≤ 0.05 *, *p* ≤ 0.01 **, *p* ≤ 0.01 ***.

**Figure 6 animals-14-00238-f006:**
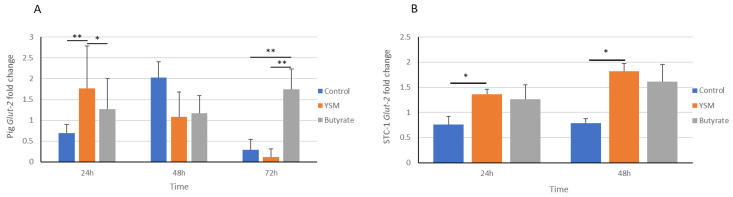
Gene expression of glucose metabolic transporter *Glut-2* in response to YSM or butyrate (pig intestinal cells, (**A**); STC-1, (**B**)). Statistical analysis was performed using one-way ANOVA and Tukey’s post hoc test (*n* = 3), *p* ≤ 0.05 *, *p* ≤ 0.01 **.

**Figure 7 animals-14-00238-f007:**
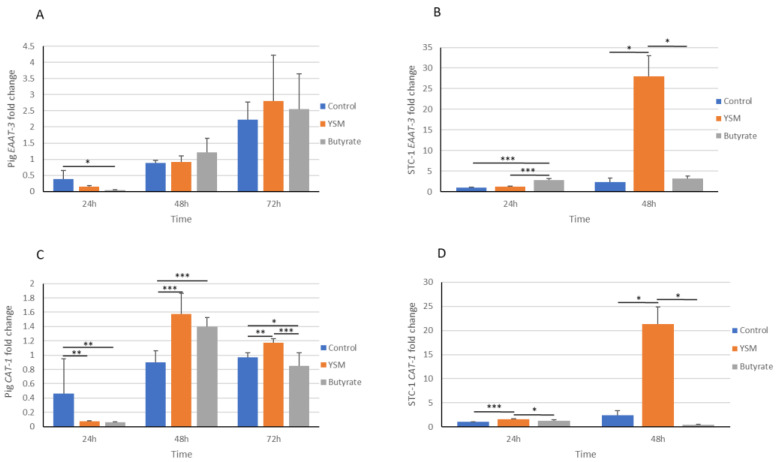
Gene expression of *EAAT-3* in response to YSM or butyrate (pig, (**A**); STC-1, (**B**)) and *CAT-1* gene expression in response to YSM or butyrate (pig, (**C**); STC-1, (**D**)). Statistical analysis was performed using one-way ANOVA and Tukey’s post hoc test (*n* = 3), *p* ≤ 0.05 *, *p* ≤ 0.01 **, *p* ≤ 0.01 ***.

**Table 1 animals-14-00238-t001:** PCR primers for pig intestinal cells and STC-1 intestinal cells.

Primer (Porcine)	F	R	Accession	Length (bp)	Reference
*PYY*	CAAGTCGTGGTAAAAGCGCC	GGGGATGTACTAAGTGGCGG	AY344365	94	[21]
*Ghrelin*	GAACTAGGCCACCAGGGAAC	GAACAGAGGTGGCTGGTCTC	AB562894	138	[21]
*Glut-2*	ATTGCGGGTCCAGTTGC	GTTCATGGTGGCCGAGTT	NM_001097417	58	[22]
*EAAT-3*	TTGGGCATTGGGCAGATCAT	TCACCATGGTCCTGAAACGG	JF521497.1	187	[23]
*CAT-1*	GCTGTCATGGCCTTCCTCTT	CTGGTACACCATGTTCGGCT	NM_001012613.1	138	[23]
*GAPDH*	AAGGAGTAAGAGCCCCTGGA	TCTGGGATGGAAACTGGAA	P00355	140	[23]
Primer (STC-1)	F	R	Accession	Length (bp)	
*PYY*	AACTGCTCTTCACAGACGAC	GTGCCCTCTTCTTAAACCAAAC	NM_145435.1	148	[24]
*Ghrelin*	ATAAGGAGAAGCCGGTGAGC	GGTCTTGGTGGTGAGGACAG	NM_001286404	71	[21]
*Glut-2*	AACCTTCCTAGCCCTGTTCT	GGCTAAGAACATTCCGGTGT	NM_031197.2	131	[21]
*EAAT-3*	ACCCACTTCACAAGGCTGTC	GCTTGTCACTGCTGGTTTGG	NM_009199	87	[21]
*CAT-1*	GCAGGTGTGAGAGGCTTTCT	CACAGCAGAGTCCACGGTAG	NM_007513	92	[21]
*ACTIN*	GGTTACAGGAAGTCCCTCAC	AAGCAATGCTGTCACCTTCC	NM_007393	106	[21]

## Data Availability

Data are contained within the article.

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
