# Peer review of "The Impact of a Proprietary Blend of Yeast Cell Wall, Short-Chain Fatty Acids, and Zinc Proteinate on Growth, Nutrient Utilisation, and Endocrine Hormone Secretion in Intestinal Cell Models"

_animals, 2024, doi:10.3390/ani14020238_

Round 1

Reviewer 1 Report

Comments and Suggestions for Authors

The manuscript titled "Impact of nutrional supplementation on growth, nutrient utilisation, and endocrine hormone secretion in intestinal cell models" . Using intestinal pig cells and STC-1 mouse intestinal neuroendocrine cell as in vitro model to determine the effects of YSM on celluar growth, metabolism and ghreline hormones.  However, the author failed to identify how YSM alters feed intake as the main goal of this manuscript.

Suggestions to Authors:

1. Introduction

a.  The supplementation of YSM contatined yeast wall, SCFAs and mineral. The introduction should give the backgrounds information why these products were chosen to generate the YSM product. Then, cititation the in vivo results indicating the beneficial effects on piglets. In end, summary the objective and goal of this study.

b. The introduction is lack of logicality.

2. Materials and Methods

c. L92, 150ppm of each product, which products? YSM?  sodium butyrate? 

d. Why use the microvascular endothelial cells?  the title is intestinal cell models.  The endothelial cell is not the target model for the author's objective.

d. L98, typo error for "foetal"

e. L97-99. The medium has DMEM+ FBS and L-glutamine, why also with Creative Bioarray L-glu and FBS?

f.  No experimental designs was given in the materials and methods. 

g. Since missing the experimental design, the protocols for measurements in the manuscript is messy.

h. The qPCR primers used in the study shall given the citation reference, product size, etc basic information.

3.Results

i. the density of cell normally presented by CCK method or proliferation index. And there are three groups? why no yeast cell wall tratments and mineral groups?

j. the Figure captions in Fig.1 shows four lines under the cures? Villgen V? or YSM? please unify the treatment group abbre.

k. What the meaning of detecting glucose and protein usage ?

l. The levels of ghrelin and PYY indicated that two different cell line were affected by YSM and Butyrate and corrdinated with the gene express. So what the underlying mechanism?  

m. Also, what the meaning of EAAT-3 gene expression ? Thoese gene expression do not help to address the objective of this manuscript.

5.Discussion

n. The reference in the discussion required update newest reference.

o. L327 The size of words is not the same.

p. The discussion did not explain the hypothesis or objective mentioned in introduction. 

r. The conclusion shall not contain the reference and required shorten.

s. The title of  the current manuscrtipt is too big or too vague. Please specific the title such as the nutritional supplementation of what?

Author Response

Thank you for you time in considering this manuscript. Please find the attached document below that addresses the issues and corrections that are now made to the manuscript based on your recommendations.

Reviewer 2 Report

Comments and Suggestions for Authors

The topic has scientific relevance. This research can contribute to understand the influence of yeast cell wall (YSM) on feed intake by examining its effect on the release of appetite-regulating hormones from both a mouse neuroendocrine cell line and primary pig intestinal cells. The study highlights the comparison of the expected benefits of increased appetite through hormonal secretions, improved intestinal function, improved metabolism and promotion of cell growth induced by YSM versus the effects of industry standard sodium butyrate. In general, the methods are appropriate for the objectives. The statistical analyses used are not the most adequate, the results must be improved. The discussion touches on the main findings and their interpretation. In general, the topics chosen for discussion are adequate and interesting. However, the weaker sides of the manuscript the presentation which is not mature enough for publication.

Specific line comments:

L102: How many flasks have you used? It is important for the statistic.

L161:

-          Why didn't you also evaluate the statistics over time?

-          Furthermore, it would be appropriate to also make statistics that allow the effects on the two cell lines to be compared. The values between the two lines are different, perhaps on one line it has more effect than on the other.

-          Have you considered the time 0?

L185: “Villigen”  à the term has never been defined previously. It would be appropriate to have the statistics in the graph, the mini table (in Figure1) is not immediately understandable.

L207 (Figure 2): Are you sure about the statistics? In the graphs of the various figures there are some points in which either a significance is indicated or it is not indicated, but visibly it does not correspond. Like in figure 2, is the 24 hour data statistically significant in the porcine cell line?

L211, 275 and 317: *** is 0.001, not 0.01 or DO you intend other?

L220 (Figure3): it would also be appropriate to include the statistical differences between butyrate and control.

Figure 4 and 5: There are no 72h in the murine cell line, in this and the following graphs. Sometimes there is control from the comparison statistics, other times it is missing.

Figure 6: butyrate is missing in the legend. As with the previous ones, I would review the statistics.

L326 : Letter size too big

L319: long-winded discussion, it is certainly interesting in terms of topics covered, but the thread of the discussion and the main point are lost. I would shorten the discussion.

L349: How do you explain the event at 72h in sows?

L369: Is "correlated" the right term? this statement should be supported by statistics.

L378-382: I suggest you review the sentence and your data.

L402: Is "Tended" the right term? this statement should be supported by statistics.

L404-406: are we sure? 0.74 to 0.744.

L426: How important is the mTOR pathway?

Author Response

(The authors gave the same response as above.)

Reviewer 3 Report

Comments and Suggestions for Authors

Reviewer Comments

Overall, a good manuscript that requires major revisions to be considered further.

Abstract:

This section needs to be modified to provide more technical details of the trial and state the results (also showing o values or means). There needs to be a clear distinction in the conclusion of the study. The authors have included too much background information in the abstract. Originally, the abstract should have provided enough information on the methods and main results with a clear conclusion. Please modify.

L24…….Define YSM, and do not start a sentence with an abbreviation.

L28…….there was no indication of how the statistics were done and authors stated only correlations and then “ elevated levels in YSM treated’’ etc……..

Introduction

The introduction is good and highlights the areas of importance for the present study.

L78…please delete the p value (not required in the introduction section).

L79…..The objective as stated appears to focus on a mechanistic approach to the study, which is not aligned with the results of the study, which is more comparative. Please modify accordingly.

M&M

To repeat this study will not be possible if this section is not well documented, unfortunately this is the case in this study. There are many missing infomation and a few are mentioned below;

L95…authors should clarify why the decision to use porcine endothelial cells, instead of a more representative intestinal cell such as the IPEC-J2 or IPEC-1 cell lines when studying a concept directly relating to the intestine of piglets.

L105-107….It is not clear how many flasks were used to represent an experimental unit and how many replications were done? Also its is not indicated how much of the YSM or butyrate were tested.

L161…what were the experimental units in each biological replicate? Or what the technical replicates were within the biological replicates.. Could you also indicate your final N=? No mention of the fixed effects and the random effects.

L161-164….You mentioned a correlation in your abstract but no detail on how this was done in the stats section.

Results

L173-174…why is the control performing better than the butyrate which was supposed to be industry standard at 72h.

For consistency for Figure1A and B, you should change the legend to reflect the treatment name YSM and not Viligen.

L192-193 ….What about the Butyrate treatment at 24h and 48h compared to the rest of the treatments?

L195….DELETE….from media at 48 h…It is a repetition.

L259-264…… appear to indicate that the ghrelin levels were reduced with YSM from 24- 48 h until 72 h in the pig intestinal cells? However, in the STC cells, ghrelin was very high at 48H, What could be the reason for this? I missed this in the discussion section.

L346-356… Is not clear what the message relays….you have repeated some results from the results section and referenced some previous work, but did not synthesize how the observed result highlight the question of the study…. It would be best if you discussed why an increase in cell densities by YSM is an important finding in light of what is known already….this is missing.

The authors did not consider highlighting the rationale for the choice of model to use in the study; there are better cell models, and the choice of endothelial cells and STC from mice must be soundly explained. This should be considered.

Overall, the discussion only repeated the results already reported and made a minimal attempt to discuss how the observed results line up with previous studies or hypotheses of the authors and eventually how this answers the study's objective. I would expect a more refined discussion section.

Comments on the Quality of English Language

Minor edits are required. A major one to consider is starting sentences with abbreviations.

Author Response

(The authors gave the same response as above.)

Round 2

Reviewer 1 Report

Comments and Suggestions for Authors

The revised manuscript basically modified as suggested.

Please revised with minor modifications.

1. Modified the fig 1's caption in figure and below the figure.

2. Shorten the conclusion.

3. The references style is not consisted in the manusctipt.

4. Combined the in vivo studies' results would be more convencible.

5. The output of the results is low quality, like fig 4 (captions in Fig 4D), re-arrenge the style of the figures.

Author Response

  1. Modified the fig 1's caption in figure and below the figure.

Response

Edited table, Figure 1. Monitoring of cell densities of pig (A) and mouse (B) intestinal cells supplemented with YSM, butyrate and un-supplemented control. Statistical analysis using One-way ANOVA, Tukeys Post-hoc test (n=3), P≤0.05 *, P≤0.01 ** and P≤0.01 *** presented in table below respective graphs 1A and 1B.

  1. Shorten the conclusion.

Response

Edited,

Here we demonstrated that YSM promoted cellular growth through greater consumption of glucose that was facilitated by glucose transport through Glut-2 in combination with higher protein usage that is potentially supported by greater appetite associated hormone secretion. Trials in pigs fed Viligen demonstrated significant improvement in ADG, FI and WG. Similarly, these in vitro findings with YSM sharethe expected outcomes observed in pigs. . Enhancing appetite in weanling piglets is vital to ensuring sustained growth and development of young pigs.

  1. The references style is not consisted in the manusctipt.

Response

References reviewed and edits made.

  1. Combined the in vivo studies' results would be more convencible.

Response

Reviewed. Efforts were made to reference between the in vivo study to this in vitro work such that if a reader wishes to find more detail on the trial they can. But the trials are looking at animal performance in terms of weight and feed intake we are looking at cellular changes which makes it difficult to directly compare although it was the authors aim to try understand the in vivo trial outcomes with in vitro cellular changes assessments.

  1. The output of the results is low quality, like fig 4 (captions in Fig 4D), re-arrenge the style of the figures.

Response

This has been changed and other graphs with minor errors are now updated, thank you for noticing these errors.

Reviewer 3 Report

Comments and Suggestions for Authors

No further comments

Author Response

Thank you again for your time and consideration of this manuscript.